

# Satellite tracking of juvenile whale sharks in the Sulu and Bohol Seas, Philippines

Gonzalo Araujo[1], Christoph A. Rohner[2], Jessica Labaja[1], Segundo J. Conales[3], Sally J. Snow[1], Ryan Murray[1], Simon J. Pierce[2] and Alessandro Ponzo[1]

[1] Large Marine Vertebrates Research Institute Philippines, Jagna, Bohol, Philippines
[2] Marine Megafauna Foundation, Truckee, CA, United States of America
[3] Tubbataha Management Office, Puerto Princesa City, Palawan, Philippines

## ABSTRACT

The whale shark *Rhincodon typus* was uplisted to 'Endangered' in the 2016 IUCN Red List due to >50% population decline, largely caused by continued exploitation in the Indo-Pacific. Though the Philippines protected the whale shark in 1998, concerns remain due to continued take in regional waters. In light of this, understanding the movements of whale sharks in the Philippines, one of the most important hotspots for the species, is vital. We tagged 17 juvenile whale sharks with towed SPOT5 tags from three general areas in the Sulu and Bohol Seas: Panaon Island in Southern Leyte, northern Mindanao, and Tubbataha Reefs Natural Park (TRNP). The sharks all remained in Philippine waters for the duration of tracking (6–126 days, mean 64). Individuals travelled 86–2,580 km (mean 887 km) at a mean horizontal speed of 15.5 ± 13.0 SD km day$^{-1}$. Whale sharks tagged in Panaon Island and Mindanao remained close to shore but still spent significant time off the shelf (>200 m). Sharks tagged at TRNP spent most of their time offshore in the Sulu Sea. Three of twelve whale sharks tagged in the Bohol Sea moved through to the Sulu Sea, whilst two others moved east through the Surigao Strait to the eastern coast of Leyte. One individual tagged at TRNP moved to northern Palawan, and subsequently to the eastern coast of Mindanao in the Pacific Ocean. Based on inferred relationships with temperature histograms, whale sharks performed most deep dives (>200 m) during the night, in contrast to results from whale sharks elsewhere. While all sharks stayed in national waters, our results highlight the high mobility of juvenile whale sharks and demonstrate their connectivity across the Sulu and Bohol Seas, highlighting the importance of the area for this endangered species.

## INTRODUCTION

The whale shark *Rhincodon typus* is the world's largest fish. The species inhabits tropical and sub-temperate waters, with seasonal aggregations across their range, usually associated with high prey availability (e.g., copepods, *Motta et al., 2010*; sergestids, *Rohner et al., 2015*; coral spawn, Holmberg et al., 2008). Most coastal aggregations are dominated by juvenile male sharks (*Norman et al., 2017*), although *Cochran et al. (2016)* reported the first known juvenile 1:1 male to female aggregation in the Red Sea. Recent observations from the

Corresponding author
Gonzalo Araujo,
g.araujo@lamave.org

Galapagos, Qatar, St Helena and Baja California (*Hearn et al., 2016*; *Robinson et al., 2017*; *Clingham et al., 2016*; *Ramírez-Macías et al., 2017*) have highlighted that adult sharks are likely to have more pelagic habitat preferences than juveniles.

Work by *Vignaud et al. (2014)* suggested that whale sharks are genetically homogenous within the Indo-Pacific. However, photographic-identification (henceforth photo-ID) data from the global online database at Wildbook for Whale Sharks (http://www.whaleshark.org) has revealed little connectivity among Indo-Pacific aggregation sites over short- to medium-term timescales (~20 years), with few demonstrated movements between non-contiguous feeding areas (*Norman et al., 2017*). While satellite telemetry studies have found whale sharks regularly cross international boundaries (*Eckert et al., 2002*; *Tyminski et al., 2015*; *Robinson et al., 2017*; *Rohner et al., 2018*), photo-ID data show that juvenile sharks, in particular, often have a high inter-annual site fidelity to specific feeding areas (*Norman et al., 2017*).

The Philippines is a global hotspot of whale shark abundance, and the associated whale shark tourism industry is important to the local economy. Whale shark tourism in the Philippines started in Donsol, Sorsogon Province, where whale sharks aggregate seasonally (Nov–Jun) to feed (*Pine, Alava & Yaptinchay, 2007*; *Quiros, 2007*). Donsol now receives up to 27,000 tourists per season and, through dedicated photo-ID, over 500 individual sharks have been identified to date (Wildbook for Whale Sharks, May 2018). Provisioning-based tourism activity arose in late 2011 at Oslob, Cebu Province, which now attracts over 182,000 tourists a year, making it the largest whale shark watching destination in the world (*Thomson et al., 2017*). Over 350 individuals have been identified at the site, where whale sharks are hand-fed daily through the year, since photo-ID started in March 2012 (Wildbook for Whale Sharks, May 2018). Around 1,000 tourists visit Panaon Island, Southern Leyte Province, per season to swim with the non-fed sharks in this area (*Araujo et al., 2017b*). Over 250 individuals have been identified at this site, typically associated with localised zooplankton blooms that occur between October and June (Wildbook for Whale Sharks, May 2018). *Araujo et al. (2014)*; *Araujo et al. (2017a)*) elaborate on the connectivity between sites in the Bohol Sea through photo-ID at dedicated study sites and through citizen science contributions, though little connectivity has been observed between these areas and Donsol (<1% of identified sharks) or Tubbataha Reefs Natural Park (TRNP) in the Sulu Sea (also <1%). Through citizen science contributions and opportunistic research effort, over 74 individuals have been identified to date at TRNP (Wildbook for Whale Sharks, May 2018).

Whale sharks were targeted by fisheries in the Philippines, before national protection in 1998 (*Alava et al., 2002*), and in Taiwan into the mid 2000s (*Hsu et al., 2007*). An estimated 1,000 whale sharks were reportedly landed yearly in Hainan Province, China, alone (*Li, Wang & Norman, 2012*). Pronounced declines in sightings and catches prompted the inclusion of the species under Appendix II of the Convention on International Trade in Endangered Species of Wild Fauna and Flora (CITES) in 2002, an 'Endangered' classification on the IUCN Red List of Threatened Species in 2016 (*Pierce & Norman, 2016*), and a listing on Appendix I of the Convention on Migratory Species (CMS) in 2017. While these conservation tools can be effective for conserving elasmobranchs (*Simpfendorfer & Dulvy, 2017*), implementation and enforcement of regulations often

vary between countries (*Li, Wang & Norman, 2012*), posing challenges for a highly mobile species like the whale shark.

International movements between Taiwan and the Philippines have been identified, through satellite telemetry and photo-ID (*Hsu et al., 2007*; *Araujo et al., 2017a*), and between the Philippines and Vietnam through satellite tracking (*Eckert et al., 2002*). The relatively close proximity of the Philippines to whale shark aggregations in adjacent countries (e.g., Cenderawasih Bay, Indonesia, *Himawan et al., 2015*), and to the major fishery in the South China Sea (*Li, Wang & Norman, 2012*), mean that understanding whale shark movements in the Philippines and Southeast Asia is essential to support effective conservation efforts on a regional level. Here, we used tethered, near-real-time satellite tags to explore the movements of juvenile whale sharks tagged in the Bohol and Sulu Seas to evaluate inter-site connectivity and identify potential anthropogenic threats that may affect sharks in this area.

## METHODS

All work was performed in collaboration with the respective Regional Offices of the Department of Environment and Natural Resources, the Department of Agriculture-Bureau of Fisheries and Aquatic Resources and the Palawan Council for Sustainable Development (Wildlife Gratuitous Permit 2017-13). All research in Tubbataha Reefs Natural Park was done in collaboration with the Tubbataha Management Office.

### Study sites

Whale sharks were tagged at three different locations (Figs. 1–3): (a: ''Panaon Island'') Panaon Island has had ongoing whale shark tourism since 2006, and dedicated research since 2013 (*Araujo et al., 2017a*). The whale shark 'season' is highly variable, with sightings reported anytime between October and June (*Araujo et al., 2017b*). (b: ''Mindanao'') Misamis Oriental and Surigao del Norte in northern Mindanao were chosen as tagging locations following reports by fisherfolk on the occurrence of whale sharks in the area. Few data are available from this region, though whale shark hunters once operated from Talisayan in Misamis Oriental and in Salay, where ~100 individuals were landed per year in the 1990's (*Alava et al., 2002*), and where *Eckert et al. (2002)* tagged two whale sharks in 1997. Both tagging sites are within the Bohol Sea, a rich ecosystem that reaches >2,000 m depth and hosts 19 species of cetaceans (*Ponzo et al., 2011*), marine turtles (*Quimpo, 2013*; *Araujo et al., 2016*), five species of mobulid rays (*Rambahiniarison et al., 2016*), and in which whale shark movements have been confirmed through photo-ID (*Araujo et al., 2014*; *Araujo et al., 2017a*). (c: "TRNP") Tubbataha Reefs Natural Park (TRNP) has been an offshore no-take marine protected area (MPA) since 1988 and a UNESCO World Heritage Site since 1993. Whale sharks were historically encountered occasionally in the park. There was a substantial increase in the number of sightings in 2014, and the site was selected as an additional tagging location.

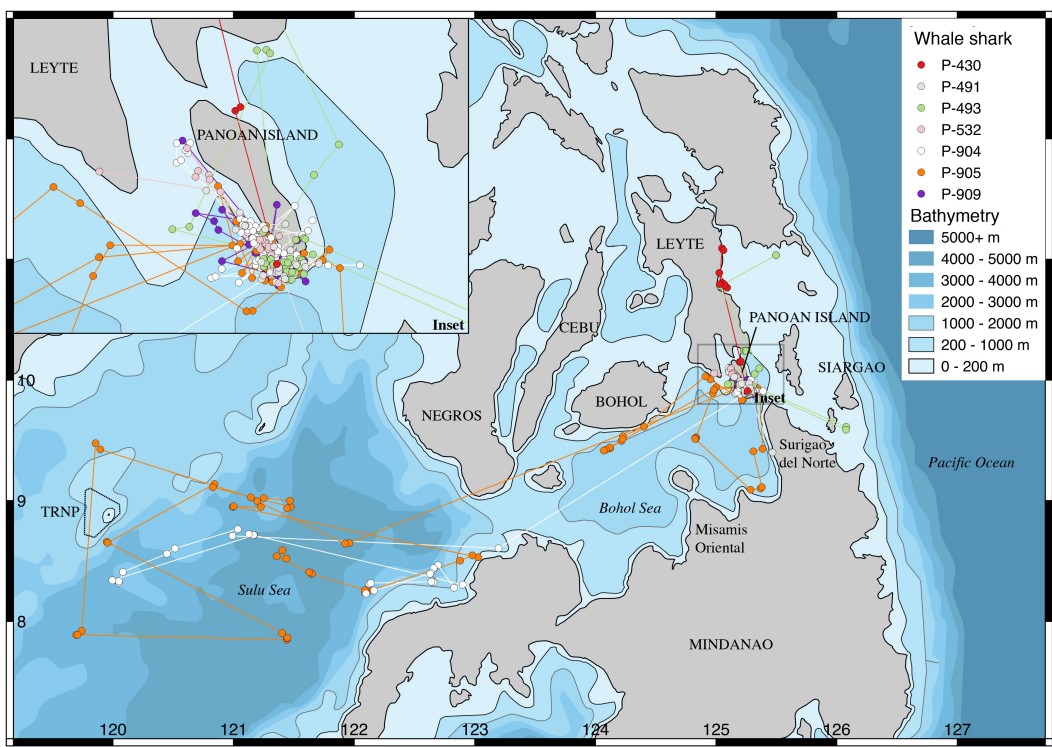

**Figure 1** Tracks of whale sharks tagged in Panaon Island, Southern Leyte.

## Photo-ID

Opportunistic whale shark surveys were conducted from small outrigger pumpboats within 1 km from shore at Panaon Island and Mindanao. Upon encountering a whale shark, a researcher entered the water and photographed the left flank of the animal, above the pectoral fin and behind the gill slits, to identify the individual (see *Arzoumanian, Holmberg & Norman, 2005*). The sex of the animal was confirmed by the presence (male) or absence (female) of claspers in the pelvic region. Size was estimated relative to an object of known length, such as swimmers or boats. Whale shark identification images were then visually checked against a site-specific database and subsequently run through the offline identification software I³S (http://www.reijns.com/i3s; *Van Tienhoven et al., 2007*) containing the same database. Newly identified individuals were uploaded onto the online database Wildbook for Whale Sharks (http://www.whaleshark.org) to assess global connectivity. Whale sharks were encountered on SCUBA at TRNP. Dive teams of two or three researchers drifted with the current at *c.* 15 m depth. Upon encountering a whale shark, the animal was photo-identified, sexed and sized as described above.

## Tagging

Wildlife Computers SPOT5 satellite tags (http://www.wildlifecomputers.com) were used to track the movement of 17 whale sharks. Tags were tethered on a 1.8 m long, 3 mm thick (240 kg breaking strain) Dyneema line. The line was attached to a titanium dart (45 × 14 × 1.3 mm), which was inserted 10–20 cm into the subdermal tissue below the
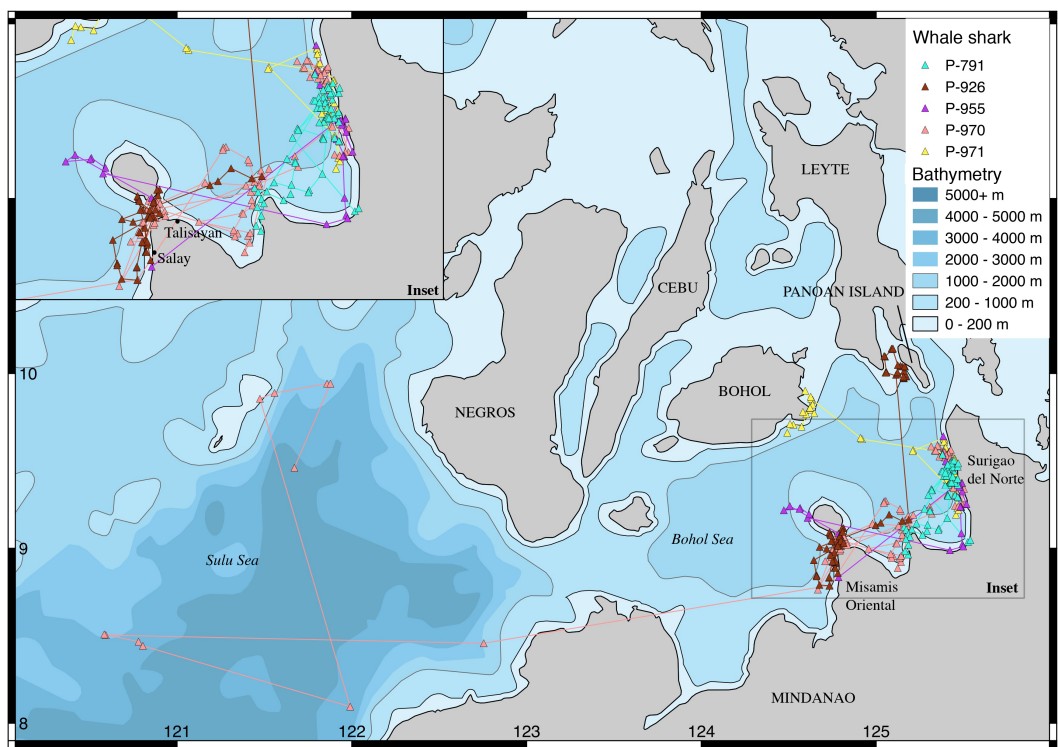

**Figure 2** Tracks of whale sharks tagged in Surigao del Norte and Misamis Oriental, Mindanao.

dorsal fin using a Hawaiian sling. The tags' positive buoyancy then allowed transmission to the ARGOS satellite system when the shark was near the surface and the tag was exposed to air. Daily transmissions were limited to 250 to maximise battery life (>180 d). Tags were deployed in Panaon Island in April and November 2015, and in Mindanao in March and April 2016 (Table 1), corresponding with known seasonality at these sites (see above). Tags at TRNP were deployed in May 2015 based on regular sightings during the tourist season (March to June). No antifouling agent was used on the tags due to a lack of availability.

## Horizontal movements

Tag location transmissions have a location class (lc: 3, 2, 1, 0, A, B, Z, in decreasing order of accuracy) associated with them. Locations transmitted before tag deployment, and after the tag detached and floated, were removed. The latter situation was detected through transmission of constant temperature histograms and early morning transmissions (00.00–03.00 h) over five consecutive days (*Hearn et al., 2013*). Locations on land (10.7% of total transmissions) were removed by extracting bathymetry data from the ETOPO dataset (*Amante & Eakins, 2009*) for each location, using the *xtractomatic* package in R (*Mendelssohn, 2017*). The bulk of remaining transmissions (69%) were from the less precise lc: B and A. The Douglas filter (*Douglas et al., 2012*) was applied to evaluate the most probable track. The filter removed unrealistic locations based on the error associated with the ARGOS location class. The filter was set to include all locations with a lc $\geq 1$ and used the maximum redundant distance (MRD) method (*Douglas et al., 2012*) with a

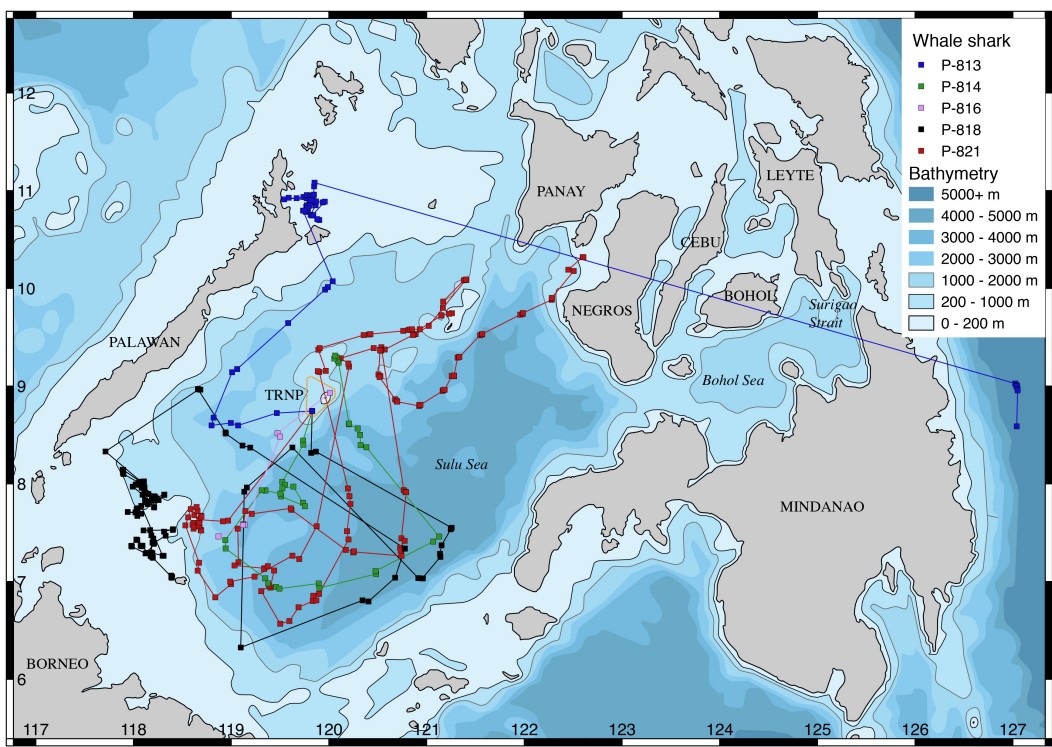

**Figure 3** Tracks of whale sharks tagged in Tubbataha Reefs Natural Park, with park boundaries in orange.

maximum redundancy of 10 km. The filter removed 158 locations—14% of the data—but kept some B and A locations that had a relatively larger error radius. The filtered tracks were used in all subsequent analyses. Tracks were plotted in QGIS (QGIS Development Team, 2017; http://qgis.osgeo.org) and track distances calculated as the sum of straight-line horizontal distances between consecutive locations, therefore representing the minimum possible distance the sharks swam. No interpolation was done.

### Time-at-temperature histograms

Tags recorded temperature in 12 pre-defined bins, <0 °C, 0–5 °C, 5–10 °C, 10-15 °C and then every 2.5 °C between 15 °C and 32.5 °C, and >32.5 °C. The temperature was measured every 10s and integrated over two time periods per day (night = 18:00–6:00; day = 6:00–18:00). These bins were used to calculate time-at-temperature (TAT) histograms. There were gaps in the TAT timeseries because tags only transmitted data on 39% of tracking days overall. Those gaps were not plotted, and therefore the $x$-axes of TAT plots are chronological but not continuous.

## RESULTS

### Photo-ID

All 5 sharks tagged at TRNP (Table 2) were new to the Philippine database at the time of tagging. Only one (P-813) was resighted at TRNP, the day after tagging, by a citizen scientist

Araujo et al. (2018), *PeerJ*, DOI 10.7717/peerj.5231

Peer J

**Table 1  Satellite tracking details for all 17 whale sharks tagged in the Sulu and Bohol Seas, Philippines.** Satellite track details, with tag number, shark ID (http://www.whaleshark.org), sex, estimated total length (TL), deployment and last transmission dates, tracking duration, number of transmitting days, overall track distance, mean speed and the number of positions per transmitting day.

| Tag | Shark | Sex | TL (cm) | Location | Deployment date | Last location | Tracking duration (d) | Transmitting days | Distance (km) | Speed (km d⁻¹) | Positions per transmitting day |
|---|---|---|---|---|---|---|---|---|---|---|---|
| 142218 | P-904 | M | 450 | Panaon Island | 17-Nov-15 | 03-Mar-16 | 108 | 60 | 1,538 | 14.2 | 2.3 |
| 142219 | P-970 | F | 650 | Mindanao | 07-Apr-16 | 23-Jun-16 | 78 | 42 | 1,661 | 21.3 | 2.2 |
| 142220 | P-905 | M | 500 | Panaon Island | 18-Nov-15 | 01-Mar-16 | 105 | 38 | 2,580 | 24.6 | 2.4 |
| 142222 | P-791 | M | 600 | Mindanao | 07-Apr-16 | 24-May-16 | 48 | 24 | 459 | 9.6 | 2.8 |
| 142224 | P-955 | F | 700 | Mindanao | 19-Mar-16 | 01-May-16 | 44 | 12 | 314 | 7.1 | 2.3 |
| 142225 | P-818 | M | 550 | TRNP | 22-May-15 | 03-Sep-15 | 105 | 45 | 2,024 | 19.3 | 2.4 |
| 142227 | P-971 | M | 450 | Mindanao | 07-Apr-16 | 29-Apr-16 | 23 | 17 | 309 | 13.4 | 2.4 |
| 142228 | P-926 | M | 500 | Mindanao | 19-Mar-16 | 17-Jun-16 | 91 | 28 | 426 | 4.7 | 2.5 |
| 142229 | P-909 | UK | 550 | Panaon Island | 18-Nov-15 | 12-Jan-16 | 56 | 20 | 178 | 3.2 | 1.9 |
| 142231 | P-821 | M | 600 | TRNP | 23-May-15 | 28-Jul-15 | 67 | 49 | 2,320 | 34.6 | 2.8 |
| 142232 | P-430 | M | 550 | Panaon Island | 10-Apr-15 | 01-Jun-15 | 53 | 9 | 149 | 2.8 | 1.0 |
| 142233 | P-493 | M | 500 | Panaon Island | 09-Apr-15 | 17-Jun-15 | 70 | 15 | 472 | 6.7 | 2.6 |
| 142235 | P-813 | F | 450 | TRNP | 17-May-15 | 14-Jul-15 | 59 | 22 | 1,493 | 25.3 | 2.6 |
| 142236 | P-814 | UK | 600 | TRNP | 17-May-15 | 01-Jun-15 | 16 | 14 | 764 | 47.8 | 2.5 |
| 142237 | P-816 | M | 550 | TRNP | 20-May-15 | 25-May-15 | 6 | 3 | 145 | 24.2 | 1.7 |
| 142238 | P-491 | M | 600 | Panaon Island | 24-Nov-15 | 28-Mar-16 | 126 | 12 | 163 | 1.3 | 2.2 |
| 142239 | P-532 | F | 600 | Panaon Island | 16-Nov-15 | 14-Dec-15 | 29 | 11 | 86 | 3.0 | 1.6 |
| | | | | | | Maximum | 126 | 60 | 2,580 | 47.8 | 2.8 |
| | | | | | | Minimum | 6 | 3 | 86 | 1.3 | 1.0 |
| | | | | | | Mean | 63.76 | 24.76 | 887.12 | 15.5 | 2.2 |
| | | | | | | S.D. | 34.96 | 16.32 | 851.90 | 13.0 | 0.5 |

**Table 2** Tagging location and resightings across different sites in the Sulu and Bohol Seas , as confirmed through photo-ID.

| Shark ID | Date 1st identified | Location 1st identified | Date of tagging | Location of tagging | Last date sighted | Location last sighted |
|---|---|---|---|---|---|---|
| P-904 | 17-Nov-15 | Panaon Island | 17-Nov-15 | Panaon Island | | |
| P-970 | 07-Apr-16 | Mindanao | 07-Apr-16 | Mindanao | | |
| P-905 | 18-Nov-15 | Panaon Island | 18-Nov-15 | Panaon Island | 20-Dec-15 | Panaon Island |
| P-791 | 25-Mar-15 | Panaon Island | 07-Apr-16 | Mindanao | | |
| P-955 | 19-Mar-16 | Mindanao | 19-Mar-16 | Mindanao | | |
| P-818 | 22-May-15 | TRNP | 22-May-15 | TRNP | | |
| P-971 | 07-Apr-16 | Mindanao | 07-Apr-16 | Mindanao | | |
| P-926 | 07-Dec-15 | Panaon Island | 19-Mar-16 | Mindanao | | |
| P-909 | 18-Nov-15 | Panaon Island | 18-Nov-15 | Panaon Island | | |
| P-821 | 23-May-15 | TRNP | 23-May-15 | TRNP | | |
| P-430 | 03-May-12 | Oslob, Cebu | 10-Apr-15 | Panaon Island | 30-Nov-17 | Panaon Island |
| P-493 | 28-Feb-13 | Panaon Island | 09-Apr-15 | Panaon Island | 02-Jan-16 | Panaon Island |
| P-813 | 17-May-15 | TRNP | 17-May-15 | TRNP | 18-May-15 | TRNP* |
| P-814 | 17-May-15 | TRNP | 17-May-15 | TRNP | | |
| P-816 | 20-May-15 | TRNP | 20-May-15 | TRNP | | |
| P-491 | 25-Feb-13 | Panaon Island | 24-Nov-15 | Panaon Island | 03-Dec-15 | Panaon Island |
| P-532 | 07-Apr-13 | Panaon Island | 16-Nov-15 | Panaon Island | 10-Jan-16 | Panaon Island |

**Notes.**
*From citizen science.

(Wildbook for Whale Sharks, February 2018). Two of the whale sharks tagged in Mindanao (P-791 and P-926) were first identified in Panaon Island in March and December 2015, respectively. No other tagged whale sharks in Mindanao were resighted. Individual P-491 was first identified in Panaon Island in February 2013 and was resighted in December 2015 (post-tagging). P-493 was first identified in Panaon Island in March 2013 and was resighted again in Panaon Island in November and December 2015, following tag detachment in June of that year. Shark P-430 was first identified in Oslob, Cebu, in March 2012. The shark was highly resident to the provisioning site (see *Araujo et al., 2014*), and was subsequently first identified at Panaon Island when it was tagged in April 2015. The shark was resighted back at Oslob in July 2016, and last seen in Panaon Island in November 2017. Individual P-532 was first identified in Panaon Island in March 2013 and tagged on November 16th 2015. The shark was resighted there again in January 2016 following tag detachment. Whale shark P-904 was tagged when first identified in November 2015 and subsequently resighted tethering the tag in December 2015. The other 2 whale sharks tagged in Panaon Island were not resighted again.

## Tagging, track duration and distances

Tagged whale sharks were all juveniles, with a mean estimated length of 5.6 m (±0.7 m S.D.) and ranging from 4.5 to 7 m (Table 1). Most of the tagged sharks were males (73%). Whale sharks at Mindanao and TRNP were not resighted post-tagging, but three individuals were resighted at Panaon Island while the tags were still attached. No obvious

tagging-related damage was observed on the animals (G Araujo, pers. obs., 2015). Tracks ranged from 6–126 days, with a mean ± SD of 64 ± 35 d. The tags transmitted locations on 39% of possible days, with a mean of 25 transmitting days per track, and a mean 2.2 transmissions per transmitting day. Whale shark track lengths ranged from 86 to 2,580 km in length, with a mean of 887 km. Mean horizontal speed was 15.5 km day$^{-1}$.

### Horizontal movements

All whale sharks stayed in the Philippines over the tracking duration. None had been subsequently identified in other countries as of February 2018. Seven sharks tagged at Panaon Island transmitted most frequently from around the tagging location (Fig. 1). Two sharks (P-904 and P-905) moved into the central Sulu Sea after having been tagged on consecutive days. Four of the Panaon Island sharks crossed the nearby Surigao Strait to the eastern coast of Leyte Island, and south of Siargao Island. Whale sharks tagged off Mindanao transmitted most frequently from the southern Bohol Sea, and none crossed the Surigao Strait (Fig. 2). One of the five sharks (P-970) swam into the Sulu Sea, while two others crossed the Bohol Sea, with P-926 swimming to Sogod Bay in Southern Leyte, and P-971 swimming to Bohol (Fig. 2). Whale sharks tagged at TRNP stayed in the Sulu Sea, with the exception of P-813 that transmitted from northern Palawan and then lost its tag in the Pacific Ocean off eastern Mindanao following 20 days of no transmissions (Fig. 3). Temperature histograms going back to six days prior to tag detachment clearly indicate that this tag was still attached to the shark while it was in transit, but the tag did not transmit a location over that period. We assume the shark swam through the Sulu and Bohol Seas into the Pacific. Sharks did not spend extended periods of time within the TRNP, with most locations transmitted from the shelf in the north of Palawan and from the shelf edge off Borneo within the Sulu Sea (Fig. 3).

### Time-at-temperature

There were 970 time-at-temperature records for all tags combined. Sharks utilised all temperature bins excepting the coldest (<0 °C). Whale sharks spent the majority (74.2%) of their time in 25–30 °C water, followed by the 30–32.5 °C (11.6%) bin (Fig. 4). Overall, 5.8% of their time was spent in <20 °C, but there were marked diurnal differences. Sharks only spent 2.1% of the daytime in colder water (<20 °C), but this increased to 9.6% at night (Fig. 4).

   Vertical movements, as inferred from TAT time-series, varied widely among individuals (Supplementary Information for all plots). Broadly, sharks spent more time at cooler temperatures when they were off the continental shelf, and during the night rather than during the day. As an example, shark P-818 (Fig. 5) was tagged in TRNP, and spent the first 4 weeks in the central Sulu Sea where it regularly dived into deeper (cooler) water, especially at night. It then spent the next three months at the continental shelf edge and on the shelf off Borneo, where ventures into cooler temperatures were infrequent (Fig. 5).

   Bathymetric depth at transmission locations ranged from 1–8, 739 m depth. 26% of all locations came from shallow shelf waters, <200 m deep. 34% of all locations were from locations over >1,000 m depth. Regional differences were observed, with 20% of

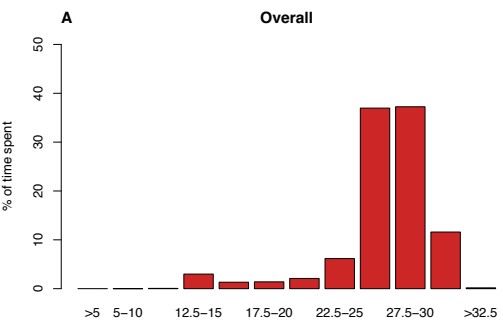
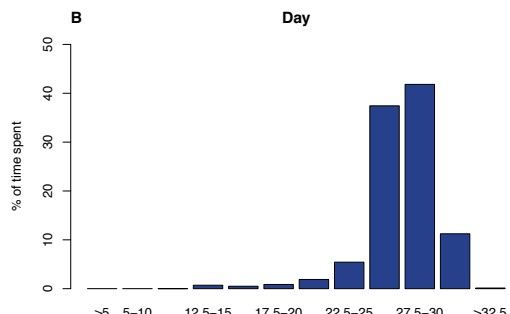

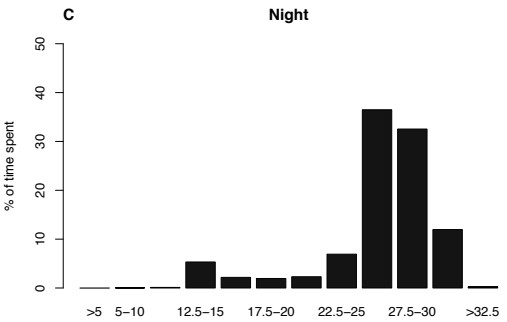

**Figure 4** Time-at-temperature histograms for all whale shark tags combined, with (A) overall results, (B) daytime observations (6 am–6 pm) and (C) nighttime observations.

locations from shelf waters for sharks tagged at Panaon Island, compared to 29% from both Mindanao and TRNP sharks.

## DISCUSSION

The tagged juvenile whale sharks all remained within the Philippines over the duration of tracking. They were, however, highly mobile, moving between the Sulu and Bohol Seas, and between the Sulu Sea and Pacific Ocean. Although juveniles had an affinity to coastal areas, they still spent 74% of their time offshore over deep water >200 m. Some whale sharks displayed both short-term site fidelity to their respective tagging areas, with transmissions received over consecutive days following tagging, and longer-term site fidelity was also demonstrated through photo-ID for some individuals. While national protection in the Philippines reduces the risk of direct anthropogenic threats to these sharks, a lack of information on female and mature sharks makes the population-level connectivity of whale sharks in Southeast Asia difficult to ascertain without the aid of other techniques, such as genetics and genomics.

### Broad-scale habitat use

Whale sharks tagged in Panaon Island spent consecutive weeks in the surrounding area, with two sharks swimming to Mindanao and/or Bohol before returning to the site. Photo-ID has previously shown that whale sharks reside a mean *c.* 27 days at Panaon Island, Southern Leyte (*Araujo et al., 2017a*) highlighting its importance as a habitat for the species. Whale

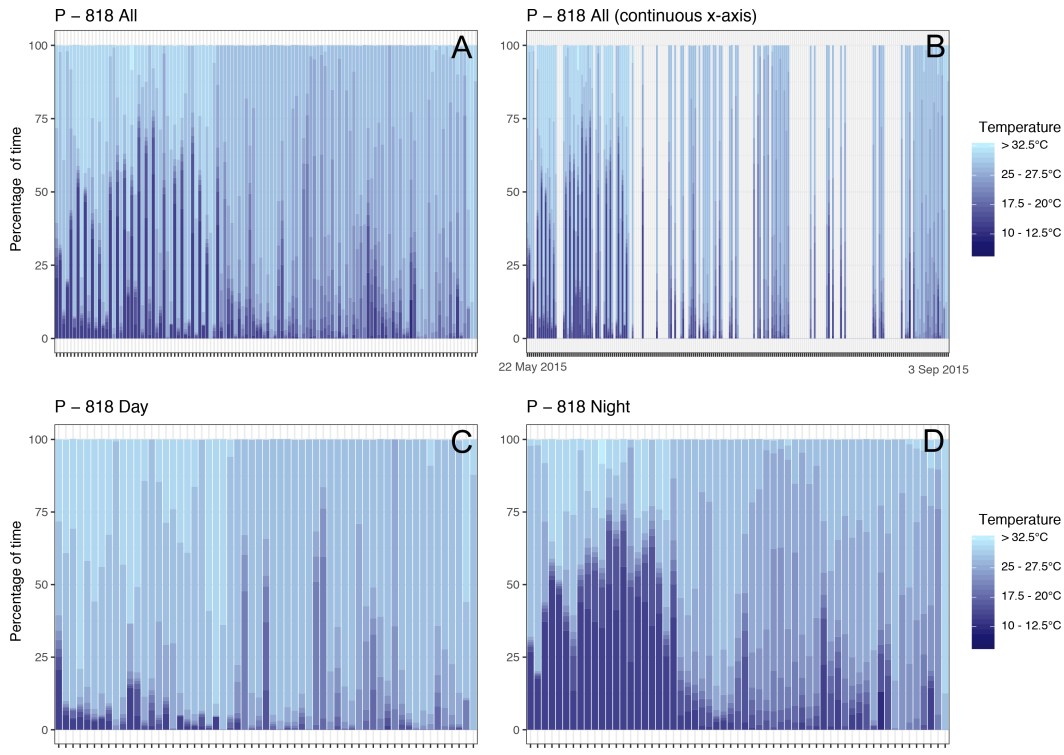

**Figure 5** **Time-at-temperature time-series for shark P-818 that was tagged in TRNP and spent its entire track within the Sulu Sea.** (A) is the entire histogram data, with a chronological *x*-axis, (B) has a continuous *x*-axis to illustrate the gaps in TAT data, (C) are all histograms from the daytime, and (D) are all histograms from the nighttime.

sharks' use of the Bohol Sea may relate to primary productivity (*Thomson et al., 2017*). Three whale sharks tagged in the Bohol Sea moved west into the Sulu Sea. A further two moved east to the eastern coast of Leyte and through the Surigao Strait. Although these movements occurred in April and May, when regional productivity typically remains relatively high (*Cabrera et al., 2011*; *Stewart et al., 2017*), the broad movement of these sharks suggests they were searching for further foraging opportunities in surrounding areas.

TRNP comprises two atolls and a smaller reef system, all of which are adjacent to deep oceanic waters. Individual P-970 (6.5 m female), originally tagged in Mindanao, transmitted from TRNP before making an almost complete change in direction of travel, swimming back towards Mindanao when the tag detached. Through photo-ID and citizen science contributions, which are high during TRNP's tourism season between March and June, it appears that whale sharks are transient to TRNP as they are rarely resighted within the same season (Wildbook for Whale Sharks, May 2018). The presence of whale sharks at TRNP could be linked to foraging—or cleaning, as has been documented in Malpelo Island, Colombia (*Quimbayo et al., 2017*)—though neither activity has been reported to date, despite the consistent presence of liveaboard dive vessels. It is plausible that TRNP is used as a navigational waypoint by whale sharks travelling through the Sulu Sea, as

previously suggested by *Acuña Marrero et al. (2014)* for Darwin Arch in the Galapagos Islands. The TRNP atolls rise from deep water (4,000 m <15 km from shore) and, together with the Cagayancillo Islands, represent some of the only land masses between Mindanao, Negros Island, and Palawan Island. Although the whale shark's ability to navigate using the earth's magnetic fields remains poorly-understood, it has been explored in other species (*Rowat & Brooks, 2012*), and it has been suggested as a possible driver of extreme dives in whale sharks (>1,000 m; *Brunnschweiller et al., 2009*; *Tyminski et al., 2015*). However, this phenomena, and the reason for their occurrence at TRNP, remain unclear.

Whale sharks spent little time (5.8%) in cooler (<20 °C) waters. The majority of their time was spent in the epipelagic zone based, on time-at-temperature (TAT) recordings. The Sulu Sea reaches a min. temperature of 9.9 °C at ~400 m, slightly cooler than the Bohol Sea's 11.6 °C (*Gordon, Sprintall & Ffield, 2011*). Whale sharks' TAT histograms show they dived into these cooler waters most frequently during the night, a reverse of the pattern observed in Mozambican whale sharks (*Rohner et al., 2018*). Dives in the upper few hundred meters are likely to relate to foraging, as whale sharks are thought to feed on meso- and bathypelagic zooplankton and fishes (*Graham, Roberts & Smart, 2006*; *Brunnschweiller et al., 2009*; *Rohner et al., 2013*). These prey species undergo daily vertical migrations, staying in dark waters at depth during the day and moving towards the surface during the night to forage (*Brierley, 2014*). Broadly sympatric mobulids capitalise on this behaviour and forage on euphausiids in the Bohol Sea during the night near the surface (*Rohner et al., 2017*). Why whale sharks appear to display a reverse pattern is unclear, and could benefit from a specific investigation through the use of archival tags capable of recording temperature and depth time series, as well as body position and acceleration, to provide more information on their behaviour.

## Ontogenetic habitat use

Recent tracking evidence from Baja California revealed preference by juveniles to coastal areas, whereas adults might have a stronger association with offshore habitats (*Ramírez-Macías et al., 2017*), supporting observations by *Ketchum, Galván-Magaña & Klimley (2013)*. Whilst this would support the general understanding as to why coastal aggregations are mostly juvenile dominated (*Rowat & Brooks, 2012*), the nature of why juveniles use offshore habitats warrants further investigation. Juveniles tagged at TRNP, located at least 150 km from the nearest major landmass, spent most of their time offshore. Contrastingly, whale sharks in Donsol, a mostly mature aggregation (53% of males are mature) and where whale shark pups were seen (*Aca & Schmidt, 2011*), are found in coastal and shallow waters seasonally, displaying strong inter-annual philopatry to the site (Wildbook for Whale Sharks, May 2018). Juveniles in the present study did spend part of their time in the open ocean, as observed elsewhere (e.g., *Robinson et al., 2017*), suggesting whale sharks use different habitats regardless of developmental stage and are perhaps more influenced by foraging opportunities not fitting the traditional 'shark nursery' concept for juveniles (*Heupel, Carlson & Simpfendorfer, 2007*), which likely occurs at the neonate stage for whale sharks (*Rowat & Brooks, 2012*).

## CONCLUSIONS AND CONSERVATION IMPLICATIONS

Satellite tagging of juvenile whale sharks in the Sulu and Bohol Seas has shed light into their short-term habitat use, over a mean of 64 days. The Sulu and Bohol Seas are an important habitat for whale sharks, with over 500 individuals identified to date in this region (Wildbook for Whale Sharks, February 2018) and where >700 individuals were harvested between 1991 and 1997 (*Alava et al., 2002*). These Seas fall under the Sulu-Sulawesi Marine Ecoregion and are central to the Coral Triangle Initiative (*Secretariat CTI, 2009*; *ADB, 2011*). Therefore, identification of threats and mitigation strategies here must be a conservation priority for the species given the historical and present population-level threats in the region, in line with the Convention on Migratory Species of the United Nations Concerted Actions for whale sharks passed in October 2017 (UNEP/CMS/Concerted Action 12.7, 2017).

This study has shown that juvenile sharks move quickly and widely through the Bohol and Sulu seas. Further work is underway to elucidate presence, seasonality and contemporary threats to whale sharks in the north Sulu Sea and southern Bohol Sea to complement the results presented herein. Targeted whale shark fisheries existed in these areas into the 1990s. Coupled with the Chinese fisheries operating in the broader region, and the established connectivity between the Philippines and Taiwan, it is imperative to monitor this population as a whole to understand if this population is in recovery, or continuing to decline. We recommend the use of longer-term satellite telemetry and molecular tools to address this key knowledge gap in Southeast Asia, and to strengthen international collaboration between and within East Asian and CTI countries.

## ACKNOWLEDGEMENTS

We would like to thank Mrs Angelique Songco and the Park Rangers for their collaboration and support while in TRNP. We would like to thank the Local Government Units and local communities of Cagayancillo, Talisayan, Malimono, Pintuyan and San Ricardo. CAR and SJP thank Marine Megafauna Foundation staff and volunteers for their assistance. We would like to extend our gratitude to Jake Levenson, Steve De Neef and the Pintuyan People's Organization "KASAKA" who helped with the overall success of this project. We thank the editor and three anonymous reviewers for their comments on the manuscript, which have strengthened our paper.

### Funding

A private trust sponsored the satellite tags and satellite time for this project. The work was partly fulfilled with support from the Rufford Foundation Small Grant 16824-1, and funding for work at TRNP was provided by 68 backers of the Indiegogo Campaign 'Expedition Shark: Discover and Protect Shark Eden' in 2015. Support for Christoph Andreas Rohner and Simon James Pierce to work on this project was provided by two

private trusts, Aqua-Firma, the Shark Foundation, and Waterlust. There was no additional external funding received for this study. The funders had no role in study design, data collection and analysis, decision to publish, or preparation of the manuscript.

## Competing Interests

The authors declare there are no competing interests.

## Author Contributions

- Gonzalo Araujo, Christoph A. Rohner, Jessica Labaja, Segundo J. Conales, Sally J. Snow, Ryan Murray Simon J. Pierce and Alessandro Ponzo conceived and designed the experiments, performed the experiments, analyzed the data, contributed reagents/materials/analysis tools, prepared figures and/or tables, authored or reviewed drafts of the paper, approved the final draft.

## Field Study Permissions

The following information was supplied relating to field study approvals (i.e., approving body and any reference numbers):

Whale shark tagging was done in partnership with the Tubbataha Reefs Natural Park Protected Area Management Board, Bureau of Fisheries and Aquatic Resources-Department of Agriculture, the Department of Environment and Natural Resources, and the Palawan Council for Sustainable Development (Wildlife Gratuitous Permit 2017-13). These agencies regulate the species and their habitats in the Philippines.

## Data Availability

The raw data are provided in a Supplemental File.

## Supplemental Information

Supplemental information for this article can be found online at http://dx.doi.org/10.7717/peerj.5231#supplemental-information.

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
