# Peer review of "Satellite tracking of juvenile whale sharks in the Sulu and Bohol Seas, Philippines"

_PeerJ, doi:10.7717/peerj.5231_

## Round 0.1 · original submission · Major Revisions

Three reviewers familiar with this field have reviewed your manuscript and their consensus decision is that the study merits publication but requires major revisions. All three reviewers have included numerous suggestions/recommendations for improvements. Please address these comments. Several reviewers commented on the need to improve grammar including standardizing on either active or passive voice. There were also numerous cases where citations were incomplete or where you can replace self-citations / personal communications /unpublished with valid citations.

I look forward to reading a substantially improved submission that addresses the three reviewers comments.

Reviewer 1 ·

Basic reporting

The manuscript suffers from a variety of grammatical issues. These include:

1) Interrupted sentence structure
2) Word misuse
3) Subjective statements
4) Out of place wording/sentences

As for references, in general there is a considerable amount of background material and context. However, the authors appear to over-cite themselves continuously via "pers obs."

The structure of the article generally has professional appeal, but there are multiple (albeit fixable) stylistic issues documented in the figures and tables.

Experimental design

The proposed work fits within the journal's Aims and Scope as it involves novel biological/environmental research. There were no major hypotheses to be tested, but a very applied question was addressed using towed-float satellite telemetry to explore movement patterns of a shark species of major economic concern. This technique was utilized with generally high technical and ethical standards (although some details were missing).

Validity of the findings

The investigation was limited based on relatively short duration tracks (generally <60 d); however, these documented important deep-shallow and inshore-offshore habitat connections that will be important for conservation and management planning moving forward. Some of the discussion involving the relationship of movements and prey availability were somewhat weak as there was no sampling of prey concomitant with shark tracking. As such, I feel the authors need to be careful not to extrapolate their findings too extensively, and recognize that there are multiple factors that may explain the patterns they observed over a relatively short-term period.

Additional comments

Please see annotated PDF for your perusal with detailed suggestions on how to improve the manuscript.

Annotated reviews are not available for download in order to protect the identity of reviewers who chose to remain anonymous.

Reviewer 2 ·

Basic reporting

The authors have added to our knowledge of whale shark dispersal and movements in the Philippines and as such the study merits publication following revisions.

The article meets structural requirements and professional english is used throughout. Restructuring of sections would improve understanding and flow and remove redundancies. The manuscript requires more primary literature cited, notably where comparable work has been conducted years previously. The authors need to state their hypotheses while further strengthening the rationale for the study, and tie their discussion and conclusions into these.

Experimental design

The research fills a geographical knowledge gap for the target species and uses commonly used tracking technology and analytical methods. However, this manuscript requires a stated research question.

Validity of the findings

Data is robust. Conclusions would need to be linked to the research question when this is formulated and included.

Additional comments

The authors have added to our knowledge of whale shark dispersal and movements in the Philippines and as such the study merits publication. The analyses are standard but the manuscript would benefit from strengthening through the removal of redundancies, restructuring of text and sections, choosing a voice, as noted below.

Specifically:

The authors may wish to articulate in their introduction greater emphasis on a strong justification for this research, also necessary to detail are the research questions that they surely used when formulating this study and the basis for deploying satellite tags (see lines 92-99). This would further help restructure and shorten the abstract to reflect answers to the research questions asked as opposed to a run-down of movement results.

Ensure identification and recognition of earliest researchers in the area e.g. LN 78 should cite Quiros 2007 and Pine et al. 2007. Instead of “Authors unpublished data” for # of sharks identified, cite Wildbook for Whale Sharks database at www.whaleshark.org.

Remove pers. Comms if you already have a citation eg LN 93. In the same vein the authors need to reduce their Authors unpublished data annotations throughout the manuscript, especially where citations exist eg Quiros 2007 noted above.

LN 92-99 setting up the premise for the study but needs to set up the justification first of why it’s a high priority for research and also note who set this priority. Then move into the research questions that will be answered by the methods used.

Methods
Streamline methods to avoid integrating elements for introduction/results/discussion. Study sites needs to focus on the sites. All other information and other studies can be appended to the introduction or discussion where it is more appropriate.

Why are the authors using two photo Identification methods (wildbook and i3s) when wildbook can provide a site specific database as well? Need to explain this further in methods, results and discussion.

Authors need to chose a voice – active or passive – and stick with it. Methods contains a mix of both.

Results
Suggest the authors follow the same sequence of the study’s elements in methods as in the results for better flow, eg. Photo ID then tagging, which would also intuitively place the sizing and sexing with the photo ID followed by tagging which usually takes place post ID. Also suggest that the tagging/distances be combined with the horizontal movements to reduce redundancies and ensure that a figure with tagging locations overlaid with movements leads the results on that section.

LN 233 Assumption that the whale shark carried the tag into the Pacific; could currents have carried it there if detached in the Sulu sea?

LN 245 Specify temperature and or ranges when mentioning “cooler”

Time at temperature: suggest creating a composite figure for all tagged sharks to identify patterns in time spent at temperatures to compare to shark 818.

Discussion
Suggest a restructuring of the discussion and conclusions to reduce redundancy within and between sections, improve flow and provide support for the importance of this study. Specifically, there is a need to set up a strong introduction and tie into your questions. Too quickly jumping into the implications at LN 262 and along the discussion subsections.

Expand citations from comparable studies that preceded Rohner to include diel vertical migration.
LN 330 remove “displaying strong inter-annual philopatry to the site (Authors, unpublished data)”

Acknowledgements
Include the research permit number and year provided.

Figures

Fig 3 – need to outline the natural park it is not showing up on the provided figures
Fig 5 include x axis timeline and add a composite for all sharks tracked to tease out diel patterns of time at temp preferences.

Reviewer 3 ·

Basic reporting

The authors are clear and unambiguous in most of the paper (there is a need for more quantitative estimates in the discussion. All citations, figures and references are well done.

Experimental design

This is a very well done study with 17 satellite tracked whale sharks. The experimental design and post hoc treatment of the data (filters) are well described.

Validity of the findings

Data is robust and conclusion are well stated. However, some opinions are given in the discussion that do not seem to be justified in the current context of the paper.

Additional comments

This is a data rich (17 spot tagged whale sharks) study that investigates the movement and residence time of an endangered and popular target species. Overall, I think the manuscript is in good shape and recommend its publication in peer J. There results that “juvenile whale sharks and their affinity not only to coastal areas, but also to offshore habitats, and reinforce our understanding of their connectivity across the Sulu and Bohol Seas, and thus highlighting the importance of the area for this endangered species” seems well supported by their data.
I have few minor issues that should be addressed.
In the abstract the claim is made “Whale sharks transited through TRNP, suggesting that these remote atolls might be used as navigational waypoints rather than as a feeding aggregation.” I don’t believe the authors can offer this fine of a mechanistic understanding. While interesting, I don’t think the data suggest that the islands are used as navigational waypoints and suggest deleting this reference.
Line 122-123 . “in 2014 there was a substantial increase in the number of sightings and thus selected as a tagging location” A little more detail here would be useful to the reader. Why was there a sudden increase (more observational opportunities, recovery of the population?)

Line 138-143, this should be in the results section.

Line 260 “Some whale sharks displayed strong site fidelity” be more quantitative what does “some” mean?

Line 263 “they still spend a substantial proportion of their time offshore” define substantial.

**In general there is a lack of quantitative details in the discussion, the data is clearly available to define these qualitative terms better.

Line 283-286. “The presence of whale sharks at TRNP could be linked to foraging – or cleaning, as has been documented in Malpelo Island, Colombia (Quimbayo et al., 2017) – though neither activity has been reported to date. Therefore, it is more likely that TRNP is used as a navigational waypoint by whale sharks traveling through the Sulu Sea”

Again, I think this is too far of a jump. It may be a possibility but the others (although not reported) seem just as likely if not more likely to me.

Line 294 “little time” please define.

Line 354-357. Whale sharks clearly move between areas and thus unsustainable tourism practices (e.g. provisioning, overcrowding, seasonal captivity) at one site have the potential of affecting the population at large. Though legislation is only the first step, it is a necessary tool to safeguard the sustainable future of this Endangered species. While I agree with the statement, I think this conclusion is not supported and beyond the scope of the paper. It is an opinion that should be further expanded into an opinion paper but doesn’t belong in the current paper.

---

## Round 0.2 · Minor Revisions

Your manuscript is greatly improved and you have addressed the comments raised by the reviewers. I found a few minor issues that need to be addressed and have marked them in the attached file. These shouldn't take more than a few minutes for you to address.

---

## Round 0.3 · accepted · Accept

Thank you for addressing the issues in the revised manuscript. Your manuscript is now satisfactory for acceptance.

#